# Effect of Maternal Hyperglycemia on Fetal Pancreatic Islet Development

**DOI:** 10.3390/biology14060728

**Published:** 2025-06-19

**Authors:** Carina Pereira Dias, Michel Raony Teixeira Paiva de Moraes, Fernanda Angela Correia Barrence, Camila Stephanie Balbino da Silva, Basilio Smuckzec, Fernanda Ortis, Telma Maria Tenório Zorn

**Affiliations:** 1Laboratory of Reproductive and Extracellular Matrix Biology, Department of Cell and Developmental Biology, Institute of Biomedical Sciences, Cidade Universitária, São Paulo 05508-000, Brazil; carina.diasws@yahoo.com.br (C.P.D.); michel_raony@hotmail.com (M.R.T.P.d.M.); fbarrence@usp.br (F.A.C.B.);; 2Tumor Biology Laboratory, Department of Cell and Developmental Biology, Institute of Biomedical Sciences, University of São Paulo, Cidade Universitária, São Paulo 05508-000, Brazil; basiliosmuczek@hotmail.com; 3Molecular Mechanism of Beta Cell Death Laboratory, Department of Cell and Developmental Biology, Institute of Biomedical Sciences, University of São Paulo, Cidade Universitária, São Paulo 05508-000, Brazil

**Keywords:** diabetes mellitus, extracellular matrix, fetal reprogramming

## Abstract

Maternal hyperglycemia during fetal development can disrupt the development of vital fetal organs, including the endocrine pancreas (islets of Langerhans). This may have lifelong consequences for energy homeostasis and glucose metabolism. In this study, we investigated the effects of maternal hyperglycemia on the deposition patterns of extracellular matrix (ECM) proteins in the fetal endocrine pancreas. The ECM plays a critical role in the proper organization and function of endocrine pancreatic cells. We observed that hyperglycemia altered the deposition pattern of key ECM proteins. These changes were accompanied by an increased presence of immature islets of Langerhans and a disrupted distribution of endocrine cell types. Additionally, we noted abnormalities in the expression of specific transcription factor proteins that regulate gene expression involved in endocrine cell proliferation and maturation. Together, these findings suggest that ECM alterations contribute to the dysfunction of the endocrine pancreas induced by maternal hyperglycemia, with potential long-term effects on the metabolic health of the offspring.

## 1. Introduction

Intrauterine growth retardation is frequently observed in murine models of pregnancy complicated by diabetes mellitus (DM) [1,2,3]. Although, in general, fetuses exhibit reduced body weight and smaller pancreatic volume, the percentage of endocrine tissue is increased, probably due to an adaptive mechanism against pancreatic insufficiency [4]. These adaptations are considered mechanisms of fetal reprogramming, a phenomenon characterized by responses to adverse intrauterine conditions that result in better fetal survival [5]. Importantly, these adaptations during intrauterine life may have long-term consequences, leading to increased susceptibility to metabolic diseases in adulthood [6,7,8,9].

DM is characterized by hyperglycemia due to beta cell dysfunction and death, in combination or not with insulin resistance [10,11,12]. DM is classified into three main types: type 1 (DM1), type 2 (DM2) and gestational diabetes (GDM) [11,12,13,14,15,16]. Chronic hyperglycemia can lead to serious complications, including reproductive disorders and impaired organogenesis in fetuses generated by hyperglycemic mothers [4,17,18,19,20,21,22,23,24,25,26,27]. Therefore, a better understanding of the mechanisms involved in this process is of pivotal importance for the improvement of maternal diabetes management.

Pancreatic development in rodents is morphologically evident from the 9th embryonic day [28]. The extracellular matrix (ECM) directly participates in this process, promoting the adhesion of pancreatic precursor cells and the interaction of its components with important signaling pathways for cell differentiation and proliferation [28]. The presence of laminins 111, 211, 221, 411 and 421 in the rodent pancreatic ECM is essential for islet cell adhesion, proliferation and normal beta cell insulin secretion [29,30]. In addition, alpha 3 integrin acts as a receptor for laminins and plays an essential role for beta cell development, regulating their survival and function [31].

Previous studies have reported that DM is a condition that can influence fetal development by altering ECM components of the intrauterine environment [32,33,34]. Our group has previously shown changes in the deposition pattern of collagens I, III and VI in the decidua, myometrium and placenta. This was accompanied by a decrease in the proliferative capacity of cells in these regions and alterations in the activity of metalloproteinases MMP2 and MMP9. In addition, there was a decrease in the weight and size of the placentas and fetuses, as well a greater number of embryonic losses and congenital malformations [32,33,34].

In this study, we investigate the effects of maternal hyperglycemia on endocrine pancreas development. To this end, we use a short-term (30–50 days) gestation model complicated by hyperglycemia, without insulin replacement and obtained by chemical induction of beta cell death [32,33,34]. Pancreatic tissue of 19-day-old mouse fetuses was used to evaluate islet morphology and cellular composition, as well as the presence of alpha 1 and gamma 1 chains of the main laminins present in the fetal pancreas and alpha 3 integrin. We also evaluate the expression of *Pdx1* and *Pax4*, transcription factors important for the differentiation and proliferation of fetal pancreatic beta cells.

## 2. Materials and Methods

### 2.1. Animals

All experiments were approved by the Institute of Biomedical Sciences Animal Ethics Committee (authorization number: Protocol 111/2016). Two-month-old Swiss mice weighing about 30 g were used. The mice were housed at constant room temperature (21 ± 1 °C), with a 12 h light/dark cycle with food and water ad libitum.

### 2.2. Induction of Diabetes and Mating Schedule

Diabetes was induced in females using a single dose of alloxan (40 mg/kg, Sigma, St. Louis, MO, USA) administered intravenously after 16 h of fasting. Females in the control group (n = 12) received an injection of 0.9% saline solution. Five days after induction, blood samples were collected from the tail vein and blood glucose was checked using a glucometer (Accu-Chek, Roche, Basel, Switzerland). Only females that presented blood glucose ≥ 400 mg/dL (n = 15) were included in the experimental group. At any point during the studies, the animals receive insulin. Thirty days after induction, females in the control and diabetes groups were mated with normoglycemic males for 30–50 days. Pregnancy was diagnosed by visualizing the vaginal plug the morning after mating, which was considered the first day of gestation or embryonic development, as previously described [32]. On day E19.0 the females were anesthetized with Avertin^®^ (25 mL.kg^−1^, Sigma, St. Louis, MO, USA), and the fetuses were collected. Fetal weight and blood glucose concentration were recorded. The crown-rump length and head and abdomen circumferences of the fetuses were measured. Following fetal decapitation, the pancreases were dissected and processed for histology and RT-qPCR analysis.

### 2.3. Tissue Collection and Histological Analyses

Fetal pancreas was freshly fixed in Methacarn (60% methanol, 30% chloroform, 10% glacial acetic acid) (Sigma, St. Louis, MO, USA) for 3 h at room temperature or 4% buffered formaldehyde solution overnight at 4 °C. The tissue was dehydrated in ethanol followed by diaphanization in xylene and embedded in paraffin wax. The tissue was sectioned at 5 μm thickness using a microtome (microergostar), the paraffin was removed at 60 °C for 1 h, and the tissue was transferred to xylene, followed by ethanol–xylene, absolute ethanol, ethanol 95% and ethanol 70%, and finally washed in distilled water. Tissue was then stained with hematoxylin and eosin for morphometrical analysis (Section 2.4) or used for immunostaining assay (Section 2.5).

### 2.4. Quantification and Assessment of Fetal Islet Size

The amount and average size of the islets within the total pancreatic tissue were analyzed by a histological approach. Pancreatic sections of 19-day fetuses stained with hematoxylin and eosin were photographed at a magnification of 200× under an optical microscope (Nikon’s Eclipse E600, Nikon, Tokyo, Japan) equipped with a Nikon DP72 CCD digital camera and analyzed with the help of the Image Pro 7.0 software Media Cybernetics. The quantity, diameter and percentage of area of each islet were evaluated by the Axion Vision s40 4.8 2.0 software (Axion Vs40 4.8 2.0 Carl Zeiss MicroImaging Gmbh, Jena, Germany).

### 2.5. Immunohistochemical Analysis of Fetal Pancreatic Tissue

Sections of fetal pancreas were deparaffinized and rinsed with 0.1 M phosphate-buffered saline solution containing 0.03% (*v*/*v*) Tween 20 (PBS-T; pH 7.2). To block endogenous peroxidase activity, sections were treated with 3.0% (*v*/*v*) hydrogen peroxide in PBS for 30 min at room temperature. Antigen retrieval was performed by heating the sections in 10 mM sodium citrate buffer pH 6.0 for 40 min at 96 °C. Non-specific antigenic sites were blocked with goat serum diluted in 10% BSA/PBS (*w*/*v*) for 90 min at room temperature. For protein detection, sections were incubated with specific primary rabbit antibodies: pan-laminin (ab1575; Abcam, diluted 1:500, Cambridge, UK), alpha 1 laminin (sc H-300 Santa Cruz H-300: SC-5582 diluted 1:50, Santa Cruz, CA, USA), gamma 1 laminin (sc5584; Santa Cruz, diluted 1:50), alpha 3 integrin (600-401 R13, Rockland diluted 1:200, Baltimore, MD, USA), anti-PCNA (ab2426 Abcam 1: 200), anti-glucagon (# A0565 1:350) and anti-insulin (# A0564 1:250). Antibodies were diluted in PBS containing 0.3% (*v*/*v*) Tween 20 and the sections were incubated overnight at 4 °C. Afterwards, the slides were incubated for 2 h with secondary antibody goat anti-rabbit IgG secondary antibody HRP-labeled (5220-0283, KPL 1:200 diluted). The reaction was revealed with 0.03% 3,3-diaminobenzidine (DAB) in PBS/0.03% containing 0.03% hydrogen peroxide, and counterstained with Harris hematoxylin. Images of the sections were obtained under 400× or 600× magnifications as described above. Evaluation of proliferative activity was performed by PCNA staining of endocrine cells and mitotic index was estimated by the percentage of PCNA-positive cells in islets counted in ten random fields. About 400 endocrine cells were counted per animal, with the positive index (PI) = n of ×100 positive cells divided by the number of cells counted at random.

### 2.6. Evaluation of Pdx1 and Pax4 Expression by RT-qPCR

Fetal pancreas (9 animals from each group) was collected and immediately stored in RNAlater^®^ solution (Sigma-Aldrich, St. Louis, MO, USA) at −20 °C. Messenger RNA (mRNA) extraction was performed by the spin column-based nucleic acid purification method using the illustra RNA spin Mini RNA Isolation Kit (GE Healthcare, Uppsala, Sweden). Total mRNA was quantified by spectrophotometry using the Epoch Microplate Spectrophotometer and Take3 Multivolume Plate (BioTek, Winooski, VT, USA). Reverse transcription was performed with random primers using 1.0 µg of total mRNA and High-Capacity cDNA Reverse Transcription (ThermoFisher Scientific, Waltham, MA, USA) following the manufacturer’s instructions. cDNA was stored at −20 °C. Quantitative RT-qPCR was performed using 200 µg of pancreas cDNA with Power SYBR^®^ Green PCR Master Mix (Applied Biosystems, Waltham, MA, USA) following kit instructions. The following primers were used: *Pdx1* (F 5′-CCAAAACCGTCGCATGAAGTG-3′; R 3′-TCTGGGTCCCAGACCCG-5′,), *Pax4* (F 5′-ACCTCATCCCAGGCCTATCT-3′; R 3′-AGGCCTCTTATGGCCAGTTT-5′) and RPL7 (F 5′-GCAGATGTACCGCACTGAGATTC-3′; R 3′-ACCTTTGGGCTTACTCCATTGATA-5′), used as the reference gene for data normalization based on its expression stability (geNorm: M-value 0.2337; NormFinder: SD: 0.0873). The assays were performed using the StepOnePlus System (Life Technologies, Waltham, MA, USA) and relative mRNA expression was calculated using the 2-ΔΔCt method [35].

### 2.7. Statistical Analysis

All data are expressed as mean ± standard error of mean. Data were compared by Student’s *t*-test with a 5% significance level. All statistical tests were performed using GraphPad Prism 5 (GraphPad Software, Boston, MA, USA).

## 3. Results

### 3.1. Maternal Hyperglycemia Induces Fetal Hyperglycemia and Decreases the Number of Viable Fetuses

Hyperglycemic females (n = 15) had a lower percentage of viable fetuses than females in the control group (n = 12) (Figure 1A). Fetal malformations were observed only in fetuses from hyperglycemic females. Hyperglycemic mothers’ fetuses (HMFs) were significantly lighter than the control group fetuses (Figure 1B). The blood glucose concentration of the HMF group was significantly higher when compared to the fetuses in the control group (Figure 1C). In addition, a pattern of body asymmetry was verified in comparison to the fetuses of the control group [18,21]. Fetuses of the HMF group were smaller compared to the control group, showing significant reductions in head and waist circumference, as well smaller head–tail length (Figure 1D–F).

### 3.2. Maternal Hyperglycemia Promotes Modification in Size and Morphology of Fetal Pancreatic Islets

The morphology and quantity of fetal pancreatic islets was compared between groups. The islets of the HMF group have immature cellular patterns, non-regular cord-like organization, as compared with the control group, where most of the islets are close to the original ducts (Figure 2A). In addition, although not statistically significant, the HMF group exhibited fewer but larger islets compared to the control group (Figure 1C and Figure 2B).

### 3.3. Maternal Hyperglycemia Decreases Pan-Laminin, Laminin Alpha 1 and Gamma 1 Chains and Increases Alpha 3 Integrin Deposition in the Fetal Pancreas

The area of positive immunostaining for pan-laminin and laminin alpha 1 and gamma 1 chain area was reduced in the islet basal membrane of the HMF group compared to the control group (Figure 3A–F). On the other hand, the area of integrin alpha 3 immunostaining was increased in the HMF group (Figure 3G,H).

### 3.4. Maternal Hyperglycemia Affects the Proliferation of Fetal Pancreas Endocrine Cells

The PCNA labeling area was reduced in islet cells of the HMF group as compared to the control group (Figure 4A,B), indicating decreased proliferation of pancreatic islet cells of the HMF group compared to that observed in the fetuses of the control group. Maternal hyperglycemia led to an increase in insulin-positive area in fetal pancreatic islets (Figure 4C,D). On the other hand, maternal hyperglycemia did not promote a significant change in the glucagon-positive labeling area (Figure 4E,F). Expression of the *Pdx1* gene was evaluated and did not show significant changes between groups (Figure 4G). Although not statistically significant, there was a tendency for higher expression of *Pax4* in the HMF group when compared to the control group (Figure 4H).

## 4. Discussion

Maternal hyperglycemia can compromise embryonic and fetal development, predisposing to complications throughout development and adulthood [2,3,4]. In our model of pregnancy complicated by severe short-term hyperglycemia, without insulin replacement, we observed an increased frequency of microsomia, body asymmetry, hyperglycemia and low fetal weight, results consistent with those previously described in models of severe DM [18,21].

Although our model does not represent a typical diabetic pregnancy, lacking spontaneous insulin resistance, as observed for gestational diabetes, and without insulin replacement, as observed for DM1, it is based on a chemically induced beta cell destruction [32,33,34]. This model allows the evaluation of the direct effect of short-term (30–50 days), yet severe, hyperglycemia on pancreas development, without the confounding influence of insulin resistance or insulin replacement. This hyperglycemic environment induced significant alterations in 19-day-old preterm fetuses’ growth, endocrine pancreas morphology and composition, both at the cellular and extracellular level.

Fetal pancreatic islets of the HMF group showed a tendency to hypertrophy, which may be associated with the increased area of insulin-labeled beta cells. Similar modifications have been observed in models of mild maternal diabetes and in adults with DM2 as a compensatory mechanism for hyperglycemia [36,37]. Although we did not measure insulin and glucagon concentration in the serum, we observed an increase in insulin-positive labeling area in HMF pancreases, suggesting an increase in beta cell functional mass. There was no change in glucagon-positive area in HMF islets, but the intensity of its labeling was still increased. However, it was not possible to reach a conclusion as to whether this hyperglycemic environmental condition was sufficient to fully compromise the functionality of beta cells, as shown by Keryvran and colleagues [2]. Of note, it is of great importance to investigate the beta cell function of the HMF group after birth and adulthood, to evaluate whether these developmental changes have lasting consequences.

When comparing the islet morphology of the control group with the HMF group, we identified that the HMF group displayed a more immature profile. Most of the islets of the HMF group showed typical signs of islets in earlier stages of maturation, with disorganized cord-like structures and still being close to the original ducts [38,39]. Although it is normal that islets complete their maturation after birth, it is expected, at this gestational age, that most of the islets are distant from their original ducts, exhibiting a more rounded shape and a well-organized cord-like arrangement [38].

Since islet cell precursor differentiation and morphogenesis is mediated by the adjacent basal membrane, with the important participation of specific laminins and integrins [40], we next assessed the impact of hyperglycemia on the islet basal membrane. The deposition of total laminins (pan-laminin) and alpha 1 and gamma 1 chains of laminins was decreased in the islet basal membrane of the HMF group. The laminin gamma 1 chain is a component of the main laminin isoforms present in the basement membranes of rodent islets (111, 211, 221, 411 and 421) which are important for the survival and maintenance of beta cells [30,41,42]. Laminin 111, composed of the alpha 1, beta 1 and gamma 1 chains, is described as the predominant laminin expressed during development with a paramount importance for the assembly of basement membranes and correct organogenesis [41,42]. Disturbances in either the pattern of assembly or destruction of these proteins increase cell necrosis, inhibit differentiation and lead to apoptosis [43].

Hyperglycemia promotes post-transductional changes in several ECM proteins [44,45]. These alterations lead to the production of hyperglycated proteins with intra- and intermolecular cross-links, compromising their conformation and rendering them more rigid and resistant to the actions of metalloproteinases, which are essential for ECM remodeling. As a result, there are irreversible changes in their deposition pattern and impaired ECM turnover [45]. As previously shown by our group, hyperglycemia decreases the expression of metalloproteinases, such as MMP2 and MMP9 [34], which in the pancreas directly participate in the migration process of endocrine cell precursors through the degradation of ECM [40]. Taken together, these results may explain the decreased pattern of laminin deposition in the fetal pancreas observed here.

On the other hand, the deposition of alpha 3 integrin was increased in fetal islets. Disruption of this integrin’s abundance may have negative effects on endocrine pancreas differentiation, patterns of survival and the proliferation of islet cells, especially beta cells [31,46]. In agreement with this, we observed a decrease in PCNA-positive islet cells in the HMF group, indicating a reduction in islet cell proliferation. It would be interesting to evaluate whether this proliferative decrease is specific for one islet cell type, considering the observed increased labeling for insulin-positive cells, with no changes in glucagon-positive cells.

We next assessed the expression of *Pdx1*, an important transcription factor for fetal pancreas development, involved in islet cell proliferation, differentiation and function, and regulating insulin synthesis [47,48]. The expression of *Pdx1*, however, was not significantly altered in the fetal pancreases of the HMF group. Of note, *Pdx1* expression was evaluated in the total pancreas rather than only in the endocrine pancreas, and thus, discrete changes in *Pdx1* expression in the endocrine cells could be overshadowed by its expression in other more abundant non-endocrine pancreatic cells.

*Pax 4* is also a key transcription factor involved in beta cell morphogenesis and is a repressor of genes important for alpha cell differentiation [49]. Here, we observed a tendency for increased expression of this transcription factor in the HMF group. Although this increase was not statistically significant, possible due to the above-mentioned mixture of cells in the pancreatic tissue, the induction of this transcription factor is observed in other hyperglycemic conditions, such as in DM2 patients, which is related to an increased beta cell proliferative profile [50]. Thus, this observation agrees with the increased population of insulin-positive cells observed here in the HMF fetal pancreas.

## 5. Conclusions

Our model of maternal hyperglycemia promotes changes in the development of the fetal endocrine pancreas in mice, differentially modifying the deposition pattern of molecules in the basement membrane within the islets, associated with a decrease in endocrine cell proliferation and an increase in β cell functional mass. These observed changes are indicative of endocrine pancreatic cell reprogramming in response to maternal hyperglycemia, potentially affecting both organ morphology and its function. Further studies are necessary to determine whether these fetal changes will be translated into long-term effects into adulthood.

## Figures and Tables

**Figure 1 biology-14-00728-f001:**
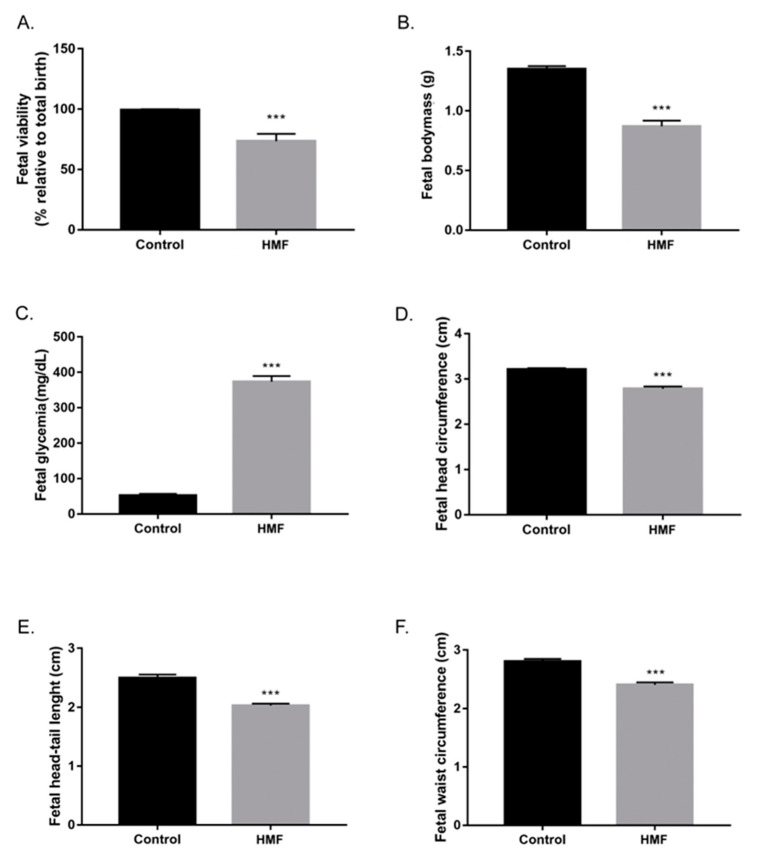
Morphometric analysis of fetuses from control and HMF groups. (**A**): Comparison between the total fetuses and number of viable fetuses. (**B**): Fetal body mass. (**C**): Fetal blood glucose. (**D**): Fetal head circumference. (**E**): Fetal head-tail length. (**F**): Fetal abdominal circumference. Data presented as mean ± standard error of the mean, *** *p* < 0.0001 vs. control group by Student’s *t*-test.

**Figure 2 biology-14-00728-f002:**
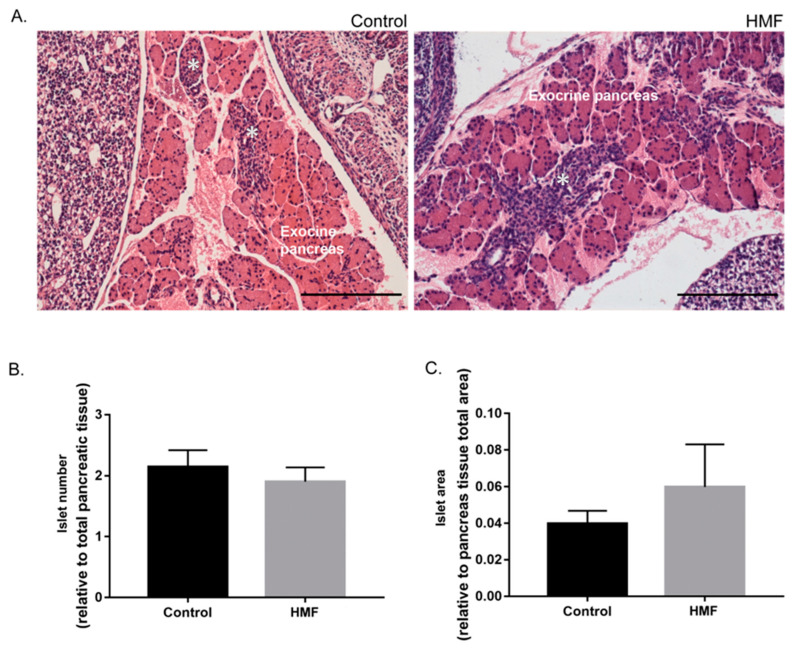
Histological analyses of the fetal pancreas. (**A**): Pancreas of control and HMF groups stained with hematoxylin and eosin. Islet of Langerhans *. Scale bar: 20.0 µm. (**B**): Total number of islets per pancreatic tissue. (**C**): Total islet area quantification. Data presented as mean ± standard error of the mean; no statistical significance by Student’s *t*-test was observed.

**Figure 3 biology-14-00728-f003:**
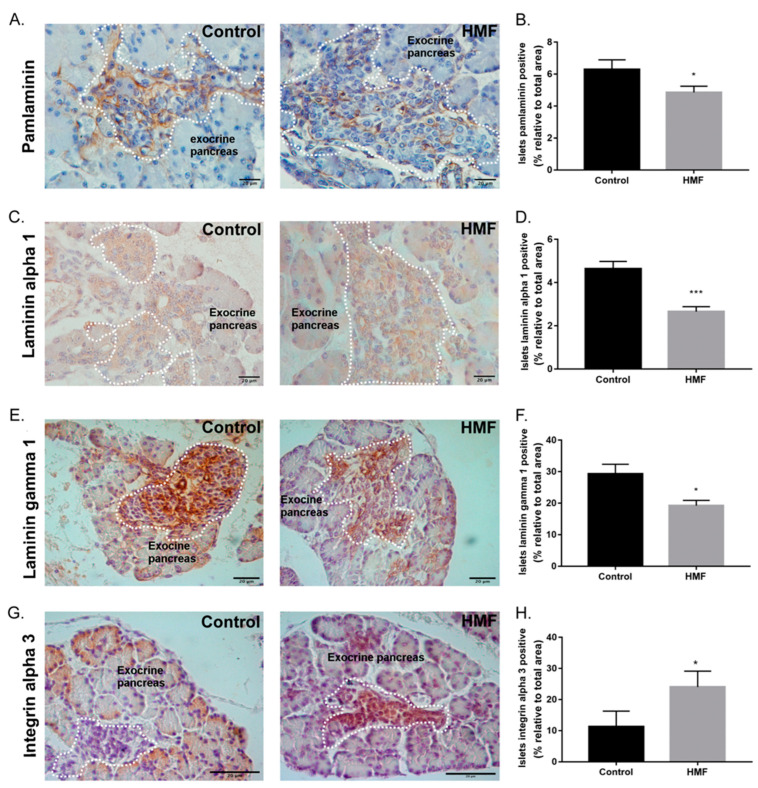
Islet of Langerhans basal membrane protein immunostaining. (**A**,**C**,**E**,**G**): Immunohistochemistry of pan-laminin, laminin alpha 1 and gamma 1 chains and alpha 3 integrin. Islet of Langerhans—dashed line. Brown staining indicates protein-of-interest staining. Scale bar: 20.0 µm. (**B**,**D**,**F**,**H**): Quantification of stained area related to total islet area for each analyzed protein. Data presented as mean ± standard error of the mean, * *p* < 0,02 and *** *p* < 0.0001 vs. control group by Student’s *t*-test.

**Figure 4 biology-14-00728-f004:**
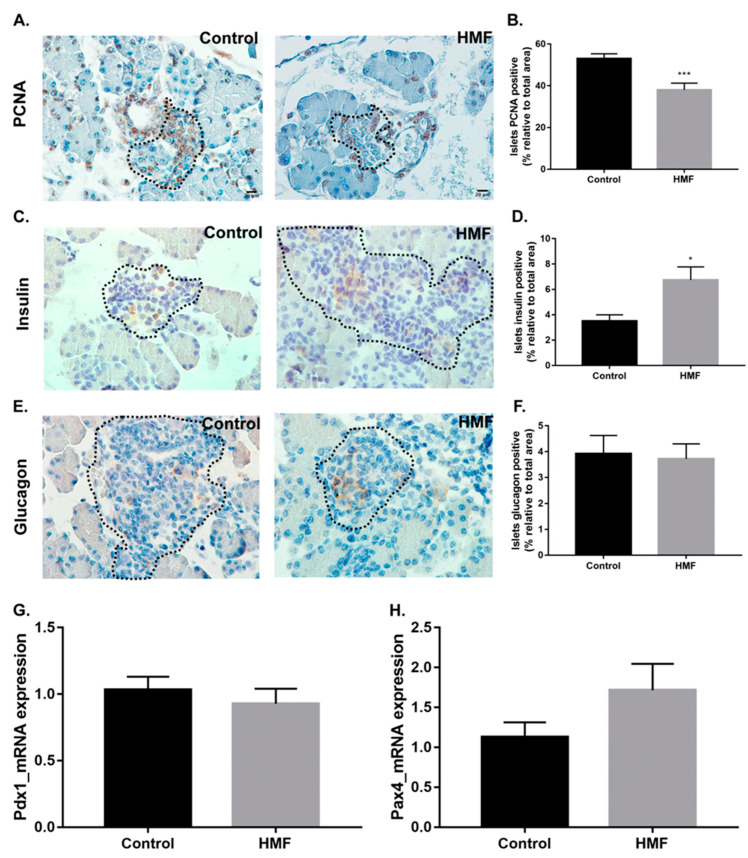
(**A**): PCNA immunostaining in pancreatic islets. Dashed line—Islet of Langerhans. Brown staining indicates positive PCNA cells. Scale bar 30.0 μm. (**B**): Quantification of endocrine cell proliferation index. The mitotic index was estimated by the percentage of positive PCNA cells counted in ten random fields. About 400 endocrine cells were counted/animal, with the positive index (PI) = n of x100 positive cells divided by the number of cells counted at random. (**C**,**E**): immunohistochemistry for insulin and glucagon, respectively. Scale bar: 20.0 µm. (**D**,**F**): Quantification of the percentages of positive stained area for insulin and glucagon, respectively. (**G**,**H**): Expression of transcription factors *Pdx1* and *Pax4*, respectively. Data presented as mean ± standard error of the mean, * *p* < 0.05; *** *p* < 0.001 by Student’s *t*-test against the control group.

## Data Availability

DOI: 10.6019/SBIAD2062.

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
