# Peer review of "Effect of Maternal Hyperglycemia on Fetal Pancreatic Islet Development"

_biology, 2025, doi:10.3390/biology14060728_

Round 1
Reviewer 1 Report
Comments and Suggestions for Authors
1) Simple Summary is written in a jumbled manner. The authors should write this section more clearly and competently
2) Abbreviations in the graphic abstract should be deciphered. A legend for the figure is required
3) [4, 17,18, 19, 20, 21, 22, 23, 24, 25, 26, 27] Excessive number of references. Key references should be left. In case of 3 or more references, they are not listed separated by commas, but presented with a hyphen.
4) Diabetes was induced and mating schedule was performed as described by Favaro R R 87 et al. 2010 [32]. It should be described in detail. Readers should be clearly aware of the methodological approach in the presented study
5) 2.3. Tissue collection and histological analyzes. The methodology should be described in detail, including the equipment used.
6) The results of PCR analysis should be confirmed by Western blotting. This is a general rule.
7) The legends of the figures should indicate the methods of statistical analysis.
8) Figure 2. The figure does not show significant differences, nor do the p values
9) The discussion should be shortened and structured.
The article should be additionally checked for grammatical and spelling errors.
Reviewer 2 Report
Comments and Suggestions for Authors
This is a very interesting and well-executed study, examining the effects of maternal hyperglycemia on fetal pancreatic development, particularly on the extracellular matrix and maturation of pancreatic islet cells. The findings contribute to important insights into the mechanisms of fetal reprogramming in diabetic pregnancies.
Study strengths.
The study addresses an important topic with translational implications for understanding developmental programming associated with diabetes.
Methodologically sound, with clear analyses of immunohistochemistry and gene expression.
The illustrations are informative and well-explained.
Ethical approval and relevant acknowledgements are appropriately indicated.
Recommendations for authors
Title and abstract:
The title contains a grammatical error. Consider revising to:
"Effect of maternal hyperglycemia on fetal pancreatic islet development"
In the abstract, there are several spelling errors ("panlaminin", "apha1") - these should be corrected.
Language and Grammar
The manuscript would benefit from a professional English review to improve clarity and grammar. Examples include:
"laminina lpha 1" → should be "laminin alpha 1"
"cordonal organelle" → should be "cord-like organelle" or "cordon organelle"
"Imperfectly developed islets of Langerhans" could be more accurately phrased as "immature islets of Langerhans"
Repetitive phrases in the discussion (e.g., "as shown", "notable") could be reduced for smoother flow.
Data Interpretation
While the observed changes in Pax4 expression are noted as not statistically significant, more emphasis should be placed on this point to avoid overinterpretation.
The conclusions will be further discussed in light of the limitations of the chemical model (compared to spontaneous gestational diabetes).
Future Directions
The authors may consider proposing postnatal follow-up studies to determine whether fetal changes translate into adult dysfunction.
A brief mention of potential clinical implications (e.g., maternal diabetes management) would improve relevance.
Recommendation
Despite minor linguistic and structural revisions required, this article presents compelling results and a solid experimental design. I recommend publication of this article with minor revisions.
It is important to note that for women who will develop diabetes in the future, who are overweight, and who donate kidneys, two important articles are:
These are women with pre-metabolic diabetes:
Comparison of pre-diabetic and non-diabetic kidney donors on blood pressure, renal biomarkers, and diabetes: a prospective cohort study. Journal of Nephrology.
Worsening of diabetic control and decline in renal function in prediabetic kidney donors compared with nondiabetic donors whose pre-kidney donation BMI was >30. In Transplantation Proceedings.
This is a very interesting and well-executed study examining the effects of maternal hyperglycemia on fetal pancreatic development, particularly the extracellular matrix and islet cell maturation.
I recommend for publication with necessary revisions.
Round 2
Reviewer 1 Report
Comments and Suggestions for Authors
I am satisfied with the authors' responses to my criticism. The authors have made the necessary changes to the article. The article can be accepted for publication in its current form.